# SSELF: ROBUST FEDERATED LEARNING AGAINST STRAGGLERS AND ADVERSARIES

## ABSTRACT

While federated learning allows efficient model training with local data at edge devices, two major issues that need to be resolved are: slow devices known as stragglers and malicious attacks launched by adversaries. While the presence of both stragglers and adversaries raises serious concerns for the deployment of practical federated learning systems, no known schemes or known combinations of schemes, to our best knowledge, effectively address these two issues at the same time. In this work, we propose Sself, a semi-synchronous entropy and loss based filtering/averaging, to tackle both stragglers and adversaries simultaneously. The stragglers are handled by exploiting different staleness (arrival delay) information when combining locally updated models during periodic global aggregation. Various adversarial attacks are tackled by utilizing a small amount of public data collected at the server in each aggregation step, to first filter out the model-poisoned devices using computed entropies, and then perform weighted averaging based on the estimated losses to combat data poisoning and backdoor attacks. A theoretical convergence bound is established to provide insights on the convergence of Sself. Extensive experimental results show that Sself outperforms various combinations of existing methods aiming to handle stragglers/adversaries.

## 1 INTRODUCTION

Large volumes of data collected at various edge devices (i.e., smart phones) are valuable resources in training machine learning models with a good accuracy. Federated learning (McMahan et al., 2017; Li et al., 2019a;b; Konečný et al., 2016) is a promising direction for large-scale learning, which enables training of a shared global model with less privacy concerns. However, current federated learning systems suffer from two major issues. First is the devices called stragglers that are considerably slower than the average, and the second is the adversaries that enforce various adversarial attacks.

Regarding the first issue, waiting for all the stragglers at each global round can significantly slow down the overall training process in a synchronous setup. To address this, an asynchronous federated learning scheme was proposed in (Xie et al., 2019a) where the global model is updated every time the server receives a local model from each device, in the order of arrivals; the global model is updated asynchronously based on the device's *staleness* $t - \tau$, the difference between the current round $t$ and the previous round $\tau$ at which the device received the global model from the server. However, among the received results at each global round, a significant portion of the results with large staleness does not help the global model in a meaningful way, potentially making the scheme ineffective. Moreover, since the model update is performed one-by-one asynchronously, the scheme in (Xie et al., 2019a) would be vulnerable to various adversarial attacks; any attempt to combine this type of asynchronous scheme with existing adversary-resilient ideas would not likely be fruitful.

There are different forms of adversarial attacks that significantly degrade the performance of current federated learning systems. First, in untargeted attacks, an attacker can poison the updated model at the devices before it is sent to the server (model update poisoning) (Blanchard et al., 2017; Lamport et al., 2019) or can poison the datasets of each device (data poisoning) (Biggio et al., 2012; Liu et al., 2017), which degrades the accuracy of the model. In targeted attacks (or backdoor attacks) (Chen et al., 2017a; Bagdasaryan et al., 2018; Sun et al., 2019), the adversaries cause the model to misclassify the targeted subtasks only, while not degrading the overall test accuracy. To resolve these issues, a robust federated averaging (RFA) scheme was recently proposed in (Pillutla et al., 2019) which utilizes the geometric median of the received results for aggregation. However, RFA tends to lose performance rapidly as the portion of adversaries exceeds a certain threshold. In this sense, RFA

is not an ideal candidate to be combined with known straggler-mitigating strategies (e.g., ignoring stragglers) where a relatively small number of devices are utilized for global aggregation; the attack ratio can be very high, significantly degrading the performance. To our knowledge, there are currently no existing methods or known combinations of ideas that can effectively handle both stragglers and adversaries at the same time, an issue that is becoming increasingly important in practical scenarios.

**Contributions.** In this paper, we propose Sself, semi-synchronous entropy and loss based filtering/averaging, a robust federated learning strategy which can tackle both stragglers and adversaries simultaneously. In the proposed idea, the straggler effects are mitigated by semi-synchronous global aggregation at the server, and in each aggregation step, the impact of adversaries are countered by a new aggregation method utilizing public data collected at the server. The details of our key ideas are as follows. Targeting the straggler issue, our strategy is to perform periodic global aggregation while allowing the results sent from stragglers to be aggregated in later rounds. The key strategy is a judicious mix of both synchronous and asynchronous approaches. At each round, as a first step, we aggregate the results that come from the same initial models (i.e., same staleness), as in the synchronous scheme. Then, we take the weighted sum of these aggregated results with different staleness, i.e., coming from different initial models, as in the asynchronous approach.

Regarding the adversarial attacks, robust aggregation is realized via entropy-based filtering and loss-weighted averaging. This can be employed at the first step of our semi-synchronous strategy described above, enabling protection against model/data poisoning and backdoor attacks. To this end, our key idea is to utilize *public IID (independent, identically distributed) data* collected at the server. We can imagine a practical scenario where the server has some global data uniformly distributed over classes, as in the setup of (Zhao et al., 2018). This is generally a reasonable setup since data centers mostly have some collected data (although they can be only a few) of the learning task. For example, different types of medical data are often open to public in various countries. Based on the public data, the server computes entropy and loss of each received model. We use the entropy of each model to filter out the devices whose models are poisoned. In addition, by taking the loss-weighted averaging of the survived models, we can protect the system against local data poisoning and backdoor attacks.

We derive a theoretical bound for Sself to ensure acceptable convergence behavior. Experimental results on different datasets show that Sself outperforms various combinations of straggler/adversary defense methods with only a small portion of public data at the server.

**Related works.** The authors of (Li et al., 2019c; Wu et al., 2019; Xie et al., 2019a) have recently tackled the straggler issue in a federated learning setup. The basic idea is to allow the devices and the server to update the models asynchronously. Especially in (Xie et al., 2019a), the authors proposed an asynchronous scheme where the global model is updated every time the server receives a local model of each device. However, a fair portion of the received models with large staleness does not help the global model in meaningful ways, potentially slowing down the convergence speed. A more critical issue here is that robust methods designed to handle adversarial attacks, such as RFA (Pillutla et al., 2019), Multi-Krum (Blanchard et al., 2017) or the presently proposed entropy/loss based idea, are hard to be implemented in conjunction with this asynchronous scheme.

To combat adversaries, various aggregation methods have been proposed in a distributed learning setup with IID data across nodes (Yin et al., 2018a;b; Chen et al., 2017b; Blanchard et al., 2017; Xie et al., 2018). The authors of (Chen et al., 2017b) suggests a geometric median based aggregation rule of the received models or the gradients. In (Yin et al., 2018a), a trimmed mean approach is proposed which removes a fraction of largest and smallest values of each element among the received results. In Multi-Krum (Blanchard et al., 2017), among $N$ workers in the system, the server tolerates $f$ Byzantine workers under the assumption of $2f + 2 < N$. Targeting federated learning with non-IID data, the recently introduced RFA method of (Pillutla et al., 2019) utilizes the geometric median of models sent from devices, similar to (Chen et al., 2017b). However, as mentioned above, these methods are ineffective when combined with a straggler-mitigation scheme, potentially degrading the performance of learning. Compared to Multi-Krum and RFA, our entropy/loss based scheme can tolerate adversaries even with a high attack ratio, showing remarkable advantages, especially when combined with straggler-mitigation schemes.

Finally, we note that the authors of (Xie et al., 2019c) considered both stragglers and adversaries but in a distributed learning setup with IID data across the nodes. Compared to these works, we target non-IID data distribution setup in a federated learning scenario.

## 2 PROPOSED FEDERATED LEARNING WITH SSELF

We consider the following federated optimization problem:

$$\mathbf{w}^* = \underset{\mathbf{w}}{\operatorname{argmin}}\, F(\mathbf{w}) = \underset{\mathbf{w}}{\operatorname{argmin}} \sum_{k=1}^{N} \frac{m_k}{m} F_k(\mathbf{w}), \tag{1}$$

where $N$ is the number of devices, $m_k$ is the number of data samples in device $k$, and $m = \sum_{k=1}^{N} m_k$ is the total number of data samples of all $N$ devices in the system. By letting $x_{k,j}$ be the $j$-th data sample in device $k$, the local loss function of device $k$, $F_k(\mathbf{w})$, is written as $F_k(\mathbf{w}) = \frac{1}{m_k} \sum_{j=1}^{m_k} \ell(\mathbf{w}; x_{k,j})$. In the following, we provide solutions aiming to solve the above problem under the existence of stragglers (subsection 2.1) and adversaries (subsection 2.2), and finally propose Sself handling both issues (subsection 2.3).

### 2.1 SEMI-SYNCHRONOUS SCHEME AGAINST STRAGGLERS

In the $t$-th global round, the server sends the current model $\mathbf{w}_t$ to $K$ devices in $S_t$ ($|S_t| = K \leq N$), which is a set of indices randomly selected from $N$ devices in the system. We let $C = K/N$ be the ratio of devices that participate at each global round. Each device in $S_t$ performs $E$ local updates with its own data and sends the updated model back to the server. In conventional federated averaging (FedAvg), the server waits until the results of all $K$ devices in $S_t$ arrive and then performs aggregation to obtain $\mathbf{w}_{t+1} = \sum_{k \in S_t} \frac{m_k}{\sum_{k \in S_t} m_k} \mathbf{w}_t(k)$, where $\mathbf{w}_t(k)$ is the model after $E$ local updates at device $k$ starting from $\mathbf{w}_t$. However, due to the effect of stragglers, waiting for all $K$ devices at the server can significantly slow down the overall training process.

In resolving this issue, our idea assumes periodic global aggregation at the server. At each global round $t$, the server transmits the current model/round $(\mathbf{w}_t, t)$ to the devices in $S_t$. Instead of waiting for all devices in $S_t$, the server aggregates the models that arrive until a fixed time deadline $T_d$ to obtain $\mathbf{w}_{t+1}$, and moves on to the next global round $t + 1$. Hence, model aggregation is performed periodically with every $T_d$. A key feature here is that we do not ignore the results sent from stragglers (not arrived by the deadline $T_d$). These results are utilized at the next global aggregation step, or even later, depending on the delay or staleness. Let $U_i^{(t)}$ be the set of devices 1) that are selected from the server at global round $t$, i.e., $U_i^{(t)} \subseteq S_t$ and 2) that successfully sent their results to the server at global round $i$ for $i \geq t$. Then, we can write $S_t = \cup_{i=t}^{\infty} U_i^{(t)}$, where $U_i^{(t)} \cap U_j^{(t)} = \emptyset$ for $i \neq j$. Here, $U_\infty^{(t)}$ can be viewed as the devices that are selected at round $t$ but failed to successfully send their results back to the server. According to these notations, the devices whose training results arrive at the server during global round $t$ belong to one of the following $t + 1$ sets: $U_t^{(0)}, U_t^{(1)}, ..., U_t^{(t)}$. Note that the result sent from device $k \in U_t^{(i)}$ is the model after $E$ local updates starting from $\mathbf{w}_i$, and we denote this model by $\mathbf{w}_i(k)$. At each round $t$, we first perform FedAvg as

$$\mathbf{v}_{t+1}^{(i)} = \sum_{k \in U_t^{(i)}} \frac{m_k}{\sum_{k \in U_t^{(i)}} m_k} \mathbf{w}_i(k) \tag{2}$$

for all $i = 0, 1, ..., t$, where $\mathbf{v}_{t+1}^{(i)}$ is the aggregated result of locally updated models (starting from $\mathbf{w}_i$) received at round $t$ with staleness $t - i + 1$. Then from $\mathbf{v}_{t+1}^{(0)}, \mathbf{v}_{t+1}^{(1)}, ..., \mathbf{v}_{t+1}^{(t)}$, we take the weighted averaging of results with different staleness to obtain $\sum_{i=0}^{t} \alpha_t(i) \mathbf{v}_{t+1}^{(i)}$. Here, $\alpha_t(i) \propto \frac{\sum_{k \in U_t^{(i)}} m_k}{(t-i+1)^c}$ is a normalized coefficient that is proportional to the number of data samples in $U_t^{(i)}$ and inversely proportional to $(t - i + 1)^c$, for a given hyperparameter $c \geq 0$. Hence, we have a larger weight for $\mathbf{v}_{t+1}^{(i)}$ with a smaller $t - i + 1$ (staleness). This is to give more weights to more recent results. Based on the weighted sum $\sum_{i=0}^{t} \alpha_t(i) \mathbf{v}_{t+1}^{(i)}$, we finally obtain $\mathbf{w}_{t+1}$ as

$$\mathbf{w}_{t+1} = (1 - \gamma) \mathbf{w}_t + \gamma \sum_{i=0}^{t} \alpha_t(i) \mathbf{v}_{t+1}^{(i)}, \tag{3}$$

where $\gamma$ combines the aggregated result with the latest global model $\mathbf{w}_t$. Now we move on to the next round $t + 1$, where the server selects $S_{t+1}$ and sends $(\mathbf{w}_{t+1}, t + 1)$ to these devices. Here, if the server knows the set of active devices (which are still performing computation), $S_{t+1}$ can be

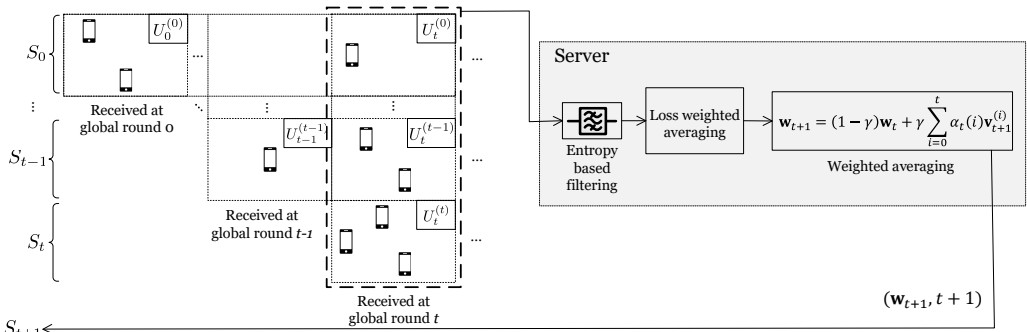

Figure 1: Overall procedure for Sself at the server. At global round $t$, the received models belong to one of the following $t + 1$ sets: $U_t^{(0)}$, $U_t^{(1)}$, ..., $U_t^{(t)}$. After entropy-based filtering, the server performs loss-weighted averaging for the results that belong to $U_t^{(i)}$ to obtain $\mathbf{v}_{t+1}^{(i)}$. Now by taking the weighted averaging of $\mathbf{v}_{t+1}^{(i)}$ for all $i = 0, 1, ..., t$ and combining with $\mathbf{w}_t$, we finally obtain $\mathbf{w}_{t+1}$ and move on to the next round $t + 1$.

constructed to be disjoint with the active devices. If not, the server randomly chooses $S_{t+1}$ among all devices in the system and the selected active devices can ignore the current request of the server. The left-hand side of Fig. 1 describes our semi-synchronous scheme.

The key characteristics of our scheme can be summarized as follows. First, by periodic global aggregation at the server, our scheme is not delayed by the effect of stragglers. Secondly, our scheme fully utilizes the results sent from stragglers in the future global rounds; we first perform federated averaging for the devices with same staleness (as in the synchronous scheme), and then take the weighted sum of these averaged results with different staleness (as in the asynchronous scheme).

## 2.2 ENTROPY AND LOSS BASED FILTERING/AVERAGING AGAINST ADVERSARIES

In this subsection, we propose entropy-based filtering and loss-weighted averaging which not only show better performance with or without attacks but also combine well with the semi-synchronous scheme compared to existing adversary-resilient aggregation methods. Our key idea is to utilize the *public IID data* collected at the server. We can imagine a practical scenario where the server (or data center) has its own data samples as in(Zhao et al., 2018), e.g., various medical data that are open to public. Using these public data at the server, we provide the following two solutions which can protect the system against model update poisoning, data poisoning and backdoor attacks.

**1) Entropy-based filtering:** Let $n_{pub}$ be the number of public data samples in the server. We also let $x_{pub,j}$ be the $j$-th sample in the server. When the server receives the locally updated models from the devices, it measures the entropy of each device $k$ by utilizing the public data as

$$E_{avg}(k) = \frac{1}{n_{pub}} \sum_{j=1}^{n_{pub}} E_{x_{pub,j}}(k), \tag{4}$$

where $E_{x_{pub,j}}(k)$ is the shannon entropy of the model of $k$-th device on the sample $x_{pub,j}$ written as $E_{x_{pub,j}}(k) = -\sum_{q=1}^{Q} P_{x_{pub,j}}^{(q)}(k) \log P_{x_{pub,j}}^{(q)}(k)$. Here, $Q$ is the number of classes of the dataset and $P_{x_{pub,j}}^{(q)}(k)$ is the probability of prediction for the $q$-th class on a sample $x_{pub,j}$, using the model of the $k$-th device. In supervised learning tasks, the model produces a high-confident prediction for the ground truth label of the trained samples and thus has a low entropy for the prediction. However, if the local model is poisoned, e.g., by reverse sign attack, the model is more likely to predict randomly for all classes and thus has a high entropy. Based on this observation, the server filters out the models that have entropy greater than some threshold value $E_{th}$. It can be seen later in Section 4 that $E_{th}$ is a hyperparameter that can be easily tuned since there is a huge gap between the entropy values of benign and adversarial devices for all datasets. Note that the above method is robust against model update poisoning even with a large portion of adversaries because it just filters out the results whose entropy is greater than $E_{th}$. This is a significant advantage compared to the median based method (Pillutla et al., 2019) whose performance is significantly degraded when the attack ratio is high.

**2) Loss-weighted averaging:** The server also measures the loss of each received model using the public data. Based on the loss values, the server then aggregates the received models as follows:

$$\mathbf{w}_{t+1} = \sum_{k \in S_t} \beta_t(k)\mathbf{w}_t(k) \quad \text{where} \quad \beta_t(k) \propto \frac{m_k}{\{F_{pub}(\mathbf{w}_t(k))\}^{\delta}} \quad \text{and} \quad \sum_{k \in S_t} \beta_t(k) = 1 \quad (5)$$

Here, $\mathbf{w}_t(k)$ is the locally updated model of the $k$-th device at global round $t$. $F_{pub}(\mathbf{w}_t(k))$ is defined as the averaged loss of $\mathbf{w}_t(k)$ computed on public data at the server, i.e., $F_{pub}(\mathbf{w}_t(k)) = \frac{1}{n_{pub}} \sum_{j=1}^{n_{pub}} \ell(\mathbf{w}_t(k); x_{pub,j})$. Finally, $\delta(\geq 0)$ in $\{F_{pub}(\cdot)\}^{\delta}$ is a parameter related to the impact of the loss on public data. We note that setting $\delta = 0$ in (5) makes our loss-weighted averaging method equal to FedAvg of (1). Under the data poisoning or backdoor attacks, the models of malicious devices would have relatively larger losses compared to others. By the definition of $\beta_t(k)$, such devices would be given a small weight and has a less impact on the next global update. By replacing the federated averaging with the above loss-weighted averaging, we are able to build a system which is robust against local data poisoning and backdoor attacks.

---

**Algorithm 1** Semi-Synchronous Entropy and Loss based Filtering/Averaging (Sself)

---

**Input:** Initialized model $\mathbf{w}_0$, **Output:** Final global model $\mathbf{w}_T$
**Algorithm at the Server**
1: **for** each global round $t = 0, 1, ..., T-1$ **do**
2:     Choose $S_t$ and send the current model and the global round $(\mathbf{w}_t, t)$ to the devices
3:     Wait for $T_d$ and then:
4:     **for** $i = 0, 1, ..., t$ **do**
5:       **for** $k \in U_t^{(i)}$ **do**
6:         $U_t^{(i)} \leftarrow U_t^{(i)} - \{k\}$, if $E_{avg}(k) > E_{th}$     // Entropy-based filtering
7:       **end for**
8:     **end for**
9:     **for** $i = 0, 1, ..., t$ **do**
10:       $\mathbf{v}_{t+1}^{(i)} = \sum_{k \in U_t^{(i)}} \beta_t(k)\mathbf{w}_i(k)$    // Loss-weighted average of results with same staleness
11:     **end for**
12:     $\mathbf{w}_{t+1} = (1-\gamma)\mathbf{w}_t + \gamma \sum_{i=0}^{t} \alpha_t(i)\mathbf{v}_{t+1}^{(i)}$   // Weighted average of results with different staleness
13: **end for**
**Algorithm at the Devices**: If device $k$ receives $(\mathbf{w}_t, t)$ from the server, it performs $E$ local updates to obtain $\mathbf{w}_t(k)$. Then each benign device $k$ sends $(\mathbf{w}_t(k), t)$ to the server, while an adversarial device transmits a poisoned model depending on the type of attack.

---

The above two methods can be easily combined to tackle model update poisoning, data poisoning and backdoor attack issues. The server first filters out the model-poisoned devices based on the entropy, and then take the loss-weighted average with the survived devices to combat data poisoning and backdoor attacks.

### 2.3 SEMI-SYNCHRONOUS ENTROPY AND LOSS BASED FILTERING/AVERAGING (SSELF)

The details of overall Sself operation are described in Algorithm 1 and Fig. 1. At global round $t$, the server chooses $S_t$ and sends $(\mathbf{w}_t, t)$ to devices. The server collects the results from the devices for a time period $T_d$, and calculates entropy $E_{avg}(k)$ and loss $F_{pub}(\mathbf{w}_t(k))$ as in (4) and (5), respectively. Based on the entropy, the server first filters out the results sent from the model poisoned devices. Then, the server aggregates the models that have the same staleness, to obtain $\mathbf{v}_{t+1}^{(i)}$ for $i = 0, 1, ..., t$. In this aggregating process, we take loss-weighted averaging as in (5) instead of conventional averaging of FedAvg, to defend the system against data poisoning or backdoor attacks. Now using $\mathbf{v}_{t+1}^{(0)}, \mathbf{v}_{t+1}^{(1)}, ..., \mathbf{v}_{t+1}^{(t)}$, we finally obtain $\mathbf{w}_{t+1}$ as in (3). Here we note that the server can compute entropy and loss whenever the model is received, i.e., based on the order of arrival. After computing entropy and loss of the last model of global round $t$, the server just needs to compute the weighted sum of the results. Hence, in practical setups where cloud servers have large enough computing powers, Sself does not cause a significant time delay at the server, compared to FedAvg. The computational complexity of Sself depends on the number of received models at each global round, and the running time for computing the entropy/loss with each model. Although direct comparison with other baselines is tricky, if we assume that the complexity of computing entropy or loss is linear to the number of model parameters as in (Xie et al., 2019b), Sself has larger complexity than that of RFA by a factor of $n_{pub}$. The additional computational complexity of Sself compared to RFA is the cost for better robustness against adversaries.

At the device-side, each device starts local model update whenever it receives $(\mathbf{w}_t, t)$ from the server. After performing $E$ local updates, device $k$ transmits $(\mathbf{w}_t(k), t)$ to the server. These processes at the server and the devices are performed in parallel and asynchronously, until the last global round ends.

## 3 Convergence Analysis

In this section, we provide insights on the convergence of Sself with the following standard assumptions in federated learning (Li et al., 2019b; Xie et al., 2019a).

**Assumption 1** *The global loss fuction $F$ defined in (1) is $\mu$-strongly convex and $L$-smooth.*

**Assumption 2** *Let $\xi_t^i(k)$ be a set of data samples that are randomly selected from the $k$-th device during the $i$-th local update at global round $t$. Then, $\mathbb{E}\|\nabla F_k(\mathbf{w}_t(k), \xi_t^i(k)) - \nabla F(\mathbf{w}_t(k))\|^2 \leq \rho_1$ holds for all $t$ and $k = 1, \ldots, N$ and $i = 1, \ldots, E$.*

**Assumption 3** *The second moments of stochastic gradients in each device is bounded, i.e., $\mathbb{E}\|\nabla F_k(\mathbf{w}_t(k), \xi_t^i(k))\|^2 \leq \rho_2$ for all $t$ and $k = 1, \ldots, N$ and $i = 1, \ldots, E$.*

We also have another assumption that describes the bounds on the error for the adversaries. Let $B_t^{(i)}$ and $M_t^{(i)}$ be the set for benign and adversarial devices of $U_t^{(i)}$ respectively, satisfying $U_t^{(i)} = B_t^{(i)} \cup M_t^{(i)}$ and $B_t^{(i)} \cap M_t^{(i)} = \emptyset$. Let

$$\Omega_t^{(i)} = \sum_{k \in M_t^{(i)}} \beta_i(k) \tag{6}$$

be the sum of loss weights for the adversarial devices in $U_t^{(i)}$. Now we have the following assumption.

**Assumption 4** *For an adversarial device $k \in M_t^{(i)}$, there exists an arbitrarily large $\Gamma$ such that $\mathbb{E}[F(\mathbf{w}_t(k)) - F(\mathbf{w}^*)] \leq \Gamma < \infty$ holds for all $i = 1, \ldots, t$.*

Based on the above assumptions, we state the following theorem which provides the convergence bound of our scheme. The proof can be found in Supplementary Material.

**Theorem 1** *Suppose Assumptions 1, 2, 3, 4 hold and the learning rate $\eta$ is set to be less than $\frac{1}{L}$. If $U_t^{(t)} \neq \emptyset$ for all $t \in \{0, 1, ..., T\}$, then Sself satisfies*

$$\mathbb{E}[F(\mathbf{w}_T) - F(\mathbf{w}^*)] \leq \nu^T[F(\mathbf{w}_0) - F(\mathbf{w}^*)] + (1 - \nu^T)C' \tag{7}$$

*where $\nu = 1 - \gamma + \gamma(1 - \eta\mu)^E$, $C' = \frac{\rho_1 + \rho_2 + 2\mu\Omega_{max}\Gamma}{2\eta\mu^2}$, $\Omega_{max} = \max\limits_{0 \leq i \leq t, 0 \leq t \leq T} \Omega_t^{(i)}$.*

We have the following important observations from Theorem 1. First, we can observe a trade-off between convergence rate $\nu^T$ and the error term $(1 - \nu^T)C'$. If we increase $\gamma$, the convergence rate improves but the error term increases as in (Xie et al., 2019a). By adjusting this $\gamma$, we can make the convergence speed faster at the beginning of training while reducing the error at the end of training. Another important observation is the impact of the adversaries. If we have a large $\Omega_{max}$ for a fixed $\nu$, it can be seen from the definition of $C'$ that we have a large error term $(1 - \nu^T)C'$. However, if the entropy-based filtering method successfully filters out the model poisoned devices, and the loss-weights $\beta_i(k)$ of the adversaries are significantly small for data poisoning and backdoor attacks, we have a small $\Omega_t^{(i)}$ (close to zero) from (6). This means that we have a significantly small $\Omega_{max}$, i.e., a small error term $(1 - \nu^T)C'$. In the next section, we show via experiments that Sself successfully combats both stragglers and adversaries simultaneously and achieves fast convergence with a small error term.

## 4 Experiments

In this section, we validate Sself on MNIST (LeCun et al., 1998), FMNIST (Xiao et al., 2017) and CIFAR-10 (Krizhevsky et al., 2009). The overall dataset is split into 60,000 training and 10,000 test samples for MNIST and FMNIST, and split into 50,000 training and 10,000 test samples for CIFAR-10. A simple convolutional neural network (CNN) with 2 convolutional layers and 2 fully connected layers is utilized for MNIST, while CNN with 2 convolutional layers and 1 fully connected layer is used for FMNIST. When training with CIFAR-10, we utilized VGG-11. We consider

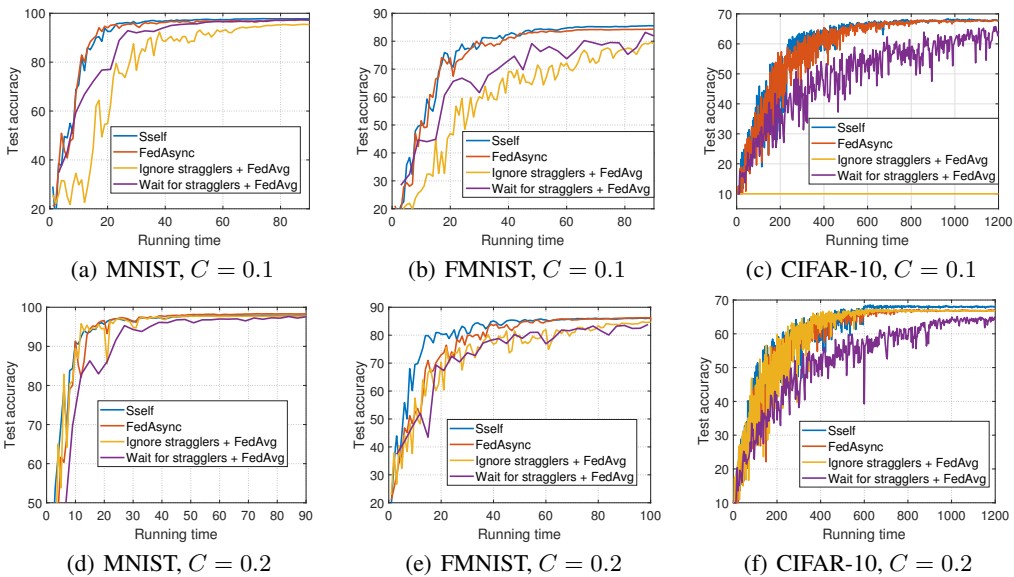

(a) MNIST, $C = 0.1$     (b) FMNIST, $C = 0.1$     (c) CIFAR-10, $C = 0.1$

(d) MNIST, $C = 0.2$     (e) FMNIST, $C = 0.2$     (f) CIFAR-10, $C = 0.2$

Figure 2: Test accuracy versus training time with only stragglers. Sself is our scheme.

$N = 100$ devices each having the same number of data samples. We randomly assigned two classes to each device to create non-IID situations. Considering the non-IID cases, we ignored the batch normalization layers when training VGG-11 with CIFAR-10. At each global round, we randomly selected a fraction $C$ of devices in the system to participate. For the proposed Sself method, we let $2\%$ of the entire training data to be the public data and performed federated training with the remaining $98\%$ of the training set. The number of local epochs at each device is set to 5 for all experiments and the local batch size is set to 10 for all experiments except for the backdoor attack. In addition, we used stochastic gradient descent and tuned hyperparameters for Sself and other comparison schemes; the details are described in Supplementary Material. Here, we emphasize that Sself outperforms existing methods even with naively chosen hyperparameters, as also shown in Supplementary Material.

**Experiments with stragglers.** To confirm the advantage of Sself, we first consider the scenario with only the stragglers. The adversarial attacks are not considered here. We compare Sself with the following methods. First is the *wait for stragglers* approach where FedAvg is applied after waiting for all the devices at each global round. The second scheme is the *ignore stragglers* approach where FedAvg is applied after waiting for a certain timeout threshold and ignore the results sent from slow devices. Finally, we consider the asynchronous scheme (FedAsync) (Xie et al., 2019a) where the global model is updated every time the result of each device arrives. For Sself and FedAsync, $\gamma$ is decayed while the learning rate is decayed in other schemes.

In Fig. 2, we plot the test accuracy versus running time on different datasets and $C$ values. For a fair comparison, the global aggregation at the server is performed with every $T_d = 1$ periodically for Sself and other comparison schemes (ignore stragglers, FedAsync). To model stragglers, each device can have delay of 0, 1, 2 which is determined independently and uniformly random. In other words, at each global round $t$, we have $S_t = U_t^{(t)} \cup U_{t+1}^{(t)} \cup U_{t+2}^{(t)}$. Our first observation from Fig. 2 is that the *ignore stragglers* scheme can lose significant data at each round and often converges to a suboptimal point with less accuracy. The *wait for stragglers* scheme requires the largest running time until convergence due to the delays caused by slow devices. Finally, it is observed that Sself performs the best, even better than the state-of-the-art FedAsync.

**Experiments with adversaries.** Next, we confirm the performance of Sself in Fig. 3 under the scenario with only the adversaries in a synchronous setup. We compare our method with geometric median-based RFA (Pillutla et al., 2019) and FedAvg under the model update/data poisoning and backdoor attacks. Comparison with the Multi-Krum is illustrated in Supplementary Material. For data poisoning attack, we conduct *label-flipping* (Biggio et al., 2012), where each label $i$ is flipped to label $i + 1$. For model update poisoning, each adversarial device takes the opposite sign of all weights and scales up 10 times before transmitting the model to the server. For both attacks, we set $C$ to 0.2 and the portion of adversarial devices is assumed to be $r = 0.2$ at each global round.

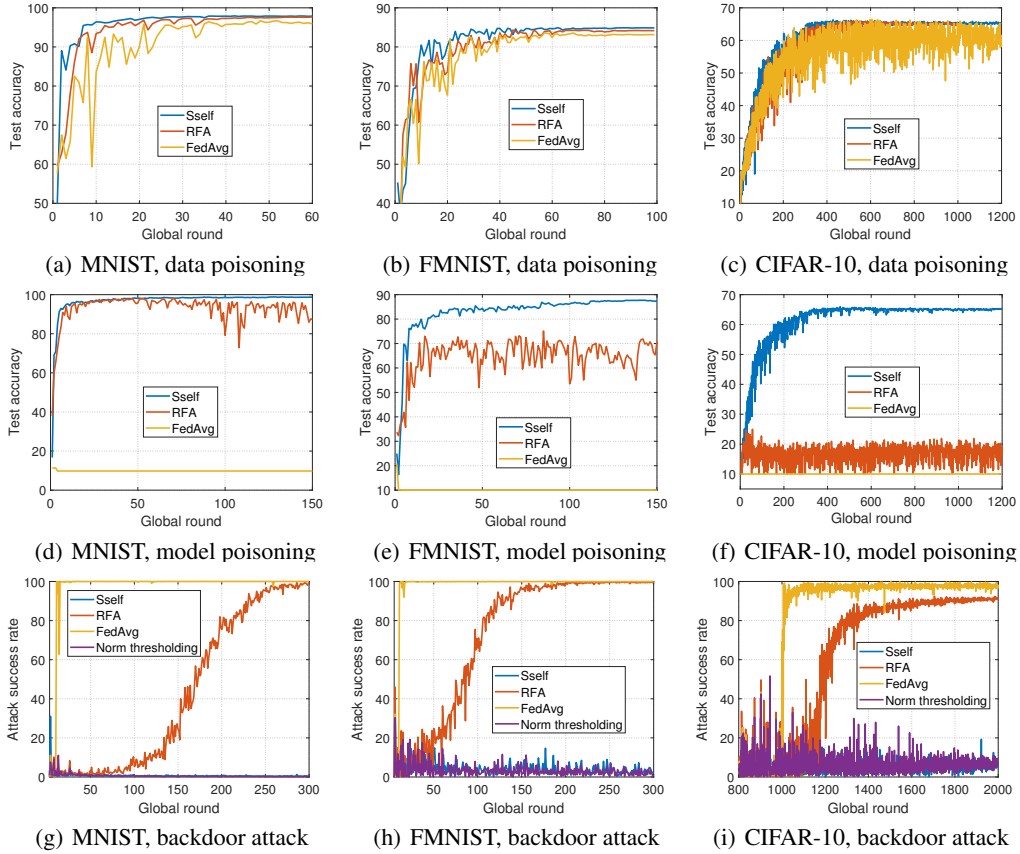

Figure 3: Performance of different schemes with only adversaries.

For the backdoor, we use the *model replacement* method (Bagdasaryan et al., 2018) in which adversarial devices transmit the scaled version of the corrupted model to replace the global model with a bad model. We conduct the *pixel-pattern backdoor attack* (Gu et al., 2017) in which the specific pixels are embedded in a fraction of images, where these images are classified as a targeted label. We embedded 12 white pixels in the top-left corner of the image and the labels of these poisoned images are set to 2. We utilize the Dirichlet distribution with parameter 0.5 for distributing training samples to $N = 100$ devices. We let $C = 0.1$, $r = 0.1$, and the local batch size is set to 64. The number of poisoned images in a batch is set to 20, and we do not decay the learning rate here. In this backdoor scenario, we additionally compare Sself with the norm-thresholding strategy (Sun et al., 2019), in which the server ignores the models with the norm greater than a pre-defined threshold. We measure the attack success rate of the backdoor task by embedding the pixel-pattern into all test samples (except data with label 2) and then comparing the predicted label with the targeted label 2. We applied backdoor attack in every round after the 10-th global round for MNIST and FMNIST, and after the 1000-th global round for CIFAR-10.

Fig. 3 shows the performance of each scheme over global round under three attack scenarios. For both data and model poisoning attacks, it can be seen that Sself shows better performance than other schemes. FedAvg does not work well on all datasets, and the performance of RFA gets worse as the dataset/neural network model become more complex. In the backdoor attack scenario, Sself and the norm-thresholding method have low attack success rates on all datasets. The other schemes cannot defend the backdoor attack having the high attack success rate as global round increases.

**Experiments with both stragglers and adversaries.** Finally in Fig. 4, we consider the setup with both stragglers and adversaries. We compare Sself with various straggler/adversary defense combinations. Comparison with the Multi-Krum is illustrated in Supplementary Material. We set $C = 0.2$, $r = 0.2$ for model/data poisoning while the results on the backdoor attack are also shown in Supplementary Material. The stragglers and adversaries are modeled as in Figs. 2 and 3, respectively. We have the following observations from Fig. 4. First, FedAsync (Xie et al., 2019a) does not perform well when combined with entropy-based filtering and loss-weighted averaging, since the model

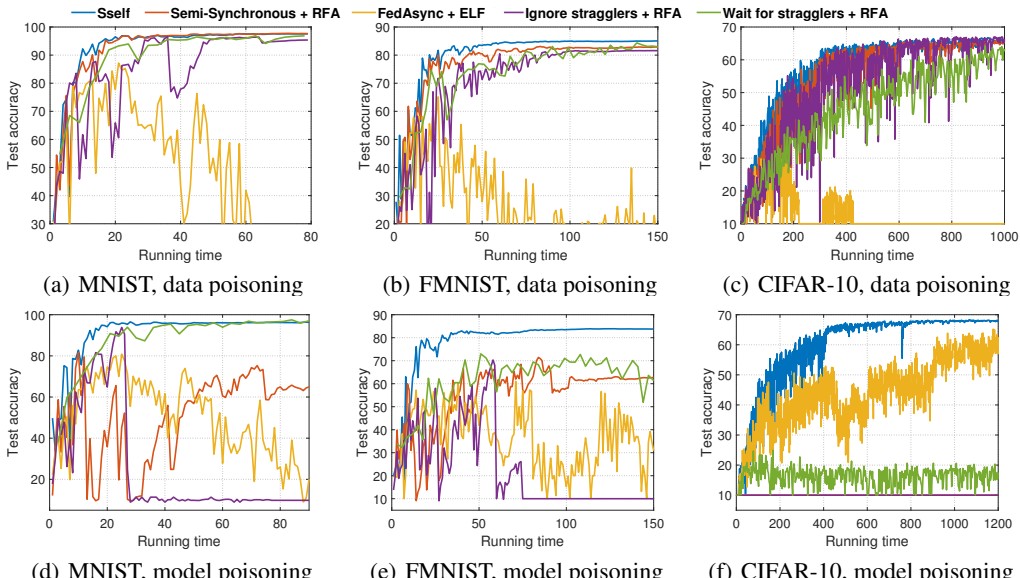

Figure 4: Performance of different schemes with both stragglers and adversaries.

update is conducted one-by-one in the order of arrivals. Due to the same issue, FedAsync cannot be combined with RFA. Our second observation is that the semi-synchronous or *ignore stragglers* method combined with RFA exhibits poor performance. The reason is that the attack ratio could often be very high (larger than $r$) for these deadline-based schemes, which degrades the performance of RFA. Compared to RFA, our entropy and loss based filtering/averaging can be applied even with a high attack ratio. It can be also seen that the *wait for stragglers* scheme combined with RFA suffers from the straggler issue. Overall, the proposed Sself algorithm performs the best, confirming significant advantages of our scheme under the existence of both stragglers and adversaries.

## 5 CONCLUSION

We proposed Sself, a robust federated learning scheme against both stragglers and adversaries. The semi-synchronous component allows the server to fully utilize the results sent from the stragglers by taking advantages of both synchronous and asynchronous elements. In each aggregation step of the semi-synchronous approach, entropy-based filtering screens out the model-poisoned devices and loss-weighted averaging reduces the impact of data poisoning and backdoor attacks. Extensive experimental results show that Sself enables fast and robust federated learning in practical scenarios with a large number of slow devices and adversaries.

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

## A  Hyperparameter Setting

### A.1  Scenario with only stragglers

The hyperparameter settings for Sself are shown in Table 1. For the schemes *ignore stragglers* and *wait for stragglers* combined with FedAvg, we decayed the learning rate during training. For the FedAsync scheme in (Xie et al., 2019a), we take a polynomial strategy with hyperparameters $a = 0.5$, $\alpha = 0.8$, and decayed $\gamma$ during training.

Table 1: Hyperparameters for Sself with only stragglers

| Dataset | $\gamma$ | $c$ | $\delta$ | $E_{th}$ | Learning rate | $\gamma$ decay |
|---|---|---|---|---|---|---|
| MNIST | 0.5 | 0.5 | 1 | 1 | 0.01 | Every 15 global epochs |
| FMNIST | 0.5 | 0.5 | 1 | 1 | 0.01 | Every 15 global epochs |
| CIFAR10 | 0.5 | 1.5 | 1 | 1 | 0.01 | Every 300 global epochs |

### A.2  Scenario with only adversaries

**Data poisoning and model update poisoning attacks:** Table 2 describes the hyperparameters for Sself with only adversaries, under data poisoning and model update poisoning attacks. For the RFA in (Pillutla et al., 2019), maximum iteration is set to 10. In this setup, the learning rate is decayed for all three schemes (Sself, RFA, FedAvg).

Table 2: Hyperparameters for Sself with only adversaries, under data and model update poisoning

| Dataset | $\gamma$ | $c$ | $\delta$ | $E_{th}$ | Learning rate | $\gamma$ decay |
|---|---|---|---|---|---|---|
| MNIST | 1 | - | 1 | 1 | 0.01 | No decay |
| FMNIST | 1 | - | 1 | 1 | 0.01 | No decay |
| CIFAR10 | 1 | - | 1 | 1 | 0.01 | No decay |

**Backdoor attack:** In this backdoor attack scenario, note that we utilized the Dirichlet distribution with parameter 0.5 for distributing training samples to N = 100 devices. Local batch size is set to 64 and the number of poisoned images is 20. In this experiment, we additionally compared our scheme with the norm-thresholding strategy (Sun et al., 2019) where the threshold value is set to 2. The hyperparameter details for Sself are shown in Table 3.

Table 3: Hyperparameters for Sself with only adversaries, under backdoor attack

| Dataset | $\gamma$ | $c$ | $\delta$ | $E_{th}$ | Learning rate | $\gamma$ decay |
|---|---|---|---|---|---|---|
| MNIST | 1 | - | 5 | 2 | 0.01 | No decay |
| FMNIST | 1 | - | 5 | 2 | 0.01 | No decay |
| CIFAR10 | 1 | - | 5 | 2 | 0.01 | No decay |

### A.3  Scenario with both stragglers and adversaries

**Data poisoning and model update poisoning attacks:** The hyperparameters for Sself are exactly the same as in Table 2.

**Backdoor attack:** The hyperparameter details are shown in Table 4.

For the comparison schemes, we considered: 1) Semi-synchronous + RFA, 2) FedAsync + ELF (entropy and loss based filtering/averaging), 3) Ignore stragglers + RFA, 4) Wait for stragglers + RFA. Each setting is set to be the same as in the previous experiments.

Table 4: Hyperparameters for Sself with both stragglers and adversaries, under backdoor attack

| Dataset | $\gamma$ | $c$ | $\delta$ | $E_{th}$ | Learning rate | $\gamma$ decay |
|---------|----------|-----|----------|----------|---------------|----------------|
| MNIST   | 0.5 | 0.5 | 5 | 2 | 0.01 | No decay |
| FMNIST  | 0.5 | 0.5 | 5 | 2 | 0.01 | No decay |
| CIFAR10 | 0.5 | 1.5 | 5 | 2 | 0.01 | Every 1000 global epochs |

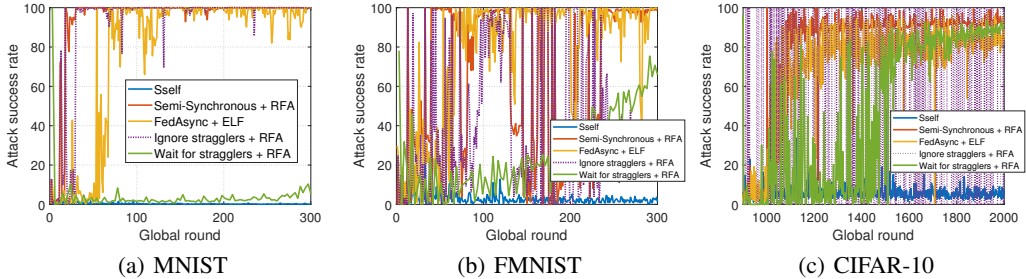

(a) MNIST      (b) FMNIST      (c) CIFAR-10

Figure B.1: Performance of different schemes with both stragglers and adversaries under backdoor attack. We set $C = 0.1$, $r = 0.1$.

# B ADDITIONAL EXPERIMENTS UNDER BACKDOOR ATTACK

## B.1 EXPERIMENTS WITH BOTH STRAGGLERS AND ADVERSARIES UNDER BACKDOOR ATTACK

Based on the hyperparameters described in Table 4, we show experimental results with both stragglers and adversaries under backdoor attack. It can be observed from Fig. B.1 that Sself successfully defends against the backdoor attack while other schemes show high attack ratios as global round increases.

## B.2 EXPERIMENTS UNDER NO-SCALED BACKDOOR ATTACK

In addition to *model replacement* backdoor attack we considered so far, we perform additional experiments under no-scaled backdoor attack (Bagdasaryan et al., 2018) where the adversarial devices do not scale the weights and only transmit the corrupted model to the server. Fig. B.2 shows the performance under no-scaled backdoor attack with only adversaries (no stragglers). It can be seen that our Sself consistently achieves low attack success rates compared to others. Since the adversaries do not scale the weights, the norm-thresholding approach cannot defend against the attack.

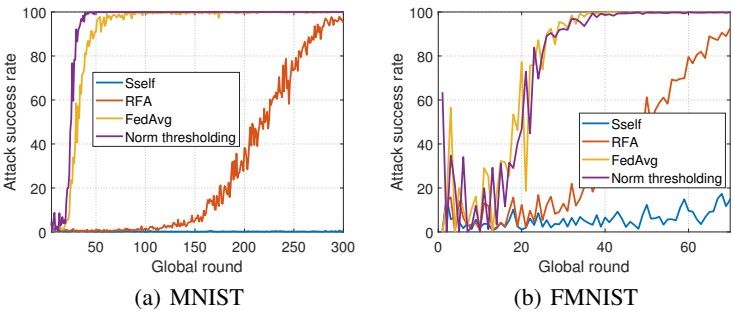

(a) MNIST      (b) FMNIST

Figure B.2: Performance comparison with no-scaled backdoor attack. We set $C = 0.1$, $r = 0.1$.

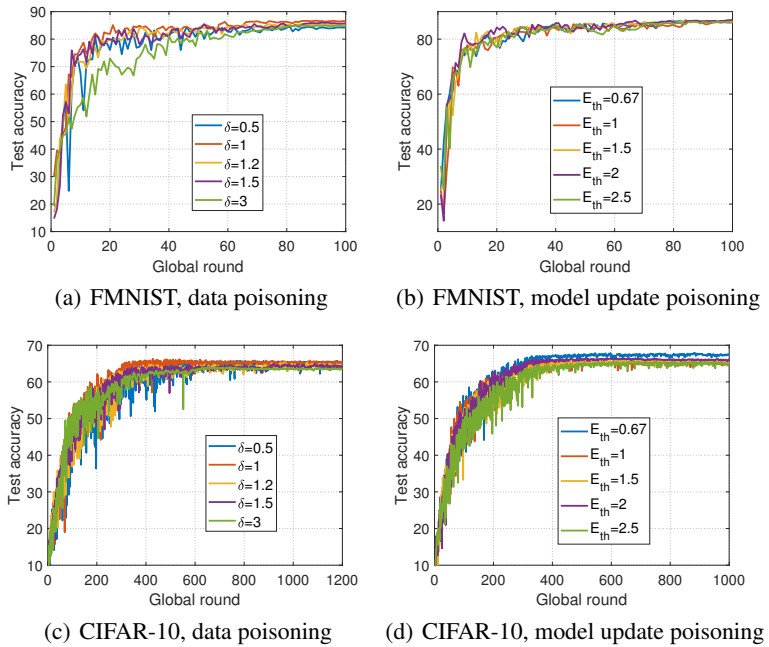

(a) FMNIST, data poisoning

(b) FMNIST, model update poisoning

(c) CIFAR-10, data poisoning

(d) CIFAR-10, model update poisoning

Figure C.1: Impact of varying hyperparameters under model update poisoning and data poisoning attacks. We set $C = 0.2$, $r = 0.2$.

## C    EXPERIMENTAL RESULTS FOR VARYING HYPERPARAMETERS

To observe the impact of hyperparameter setting, we performed additional experiments with various $\delta$ and $E_{th}$ values, the key hyperparameters of Sself. The results are shown in Fig. C.1 with only adversaries. We performed data poisoning attack for varying $\delta$ and model update poisoning attack for varying $E_{th}$. It can be seen that our scheme still performs well (better than RFA), even with naively chosen hyperparameters, confirming the advantage of Sself in terms of reducing the overhead associated with hyperparameter tuning.

## D    PERFORMANCE COMPARISON WITH MULTI-KRUM

While we compared Sself with RFA in our main manuscript, here we compare our scheme with *Multi-Krum* (Blanchard et al., 2017) which is a Byzantine-resilient aggregation method targeting conventional distributed learning setup with IID data across nodes. In Multi-Krum, among $N$ workers in the system, the server tolerates $f$ Byzantine workers under the assumption of $2f + 2 < N$. After filtering $f$ worker nodes based on squared-distances, the server chooses $N$ workers among $N - f$ remaining workers with the best scores and aggregates them. We set $M = N - f$ for comparing our scheme with Multi-Krum.

Fig. C.2 compares Sself with Multi-Krum under model update poisoning. We first observe Figs. 2(a) and 2(b) which show the results with only adversaries. It can be seen that if the number of adversaries exceed $f$, the performance of Multi-Krum significantly decreases. Compared to Multi-Krum, the proposed Sself method can filter out the poisoned devices and then take the weighted sum of the survived results even when the portion of adversaries is high. Figs. 2(c) and 2(d) show the results under the existence of both stragglers and adversaries, under model update poisoning attack. The parameter $f$ of Multi-Krum is set to the maximum value satisfying $2f + 2 < N$, where $N$ depends on the number of received results for both semi-synchronous and *ignore stragglers* approaches. However, even when we set $f$ to the maximum value, the number of adversaries can still exceed $f$, which degrades the performance of Multi-Krum combined with semi-synchronous and *ignore stragglers* approaches. Obviously, Multi-Krum can be combined with the *wait for stragglers* approach by setting

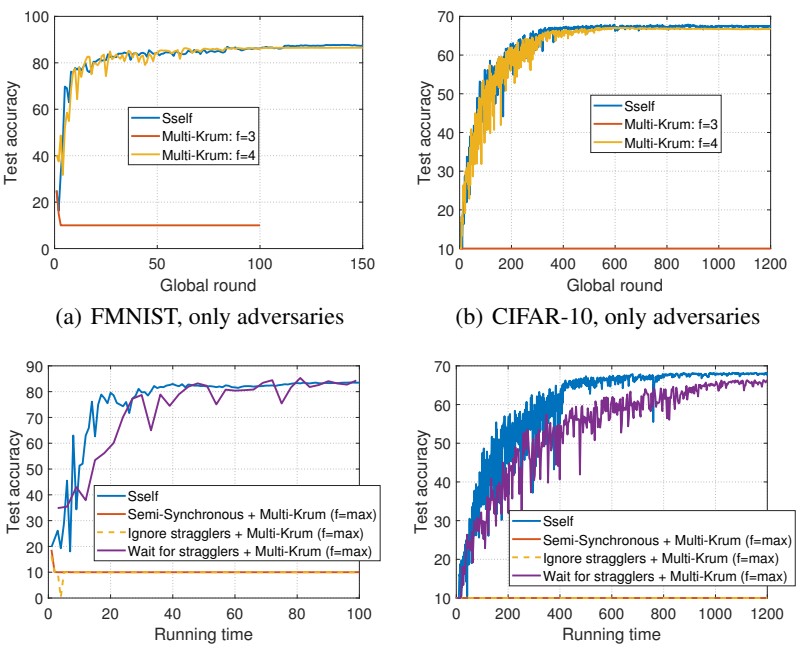

(a) FMNIST, only adversaries  (b) CIFAR-10, only adversaries

(c) FMNIST, both stragglers/adversaries  (d) CIFAR-10, both stragglers/adversaries

Figure C.2: Performance comparison with Multi-Krum under model update poisoning. We set $C = 0.2$ and $r = 0.2$.

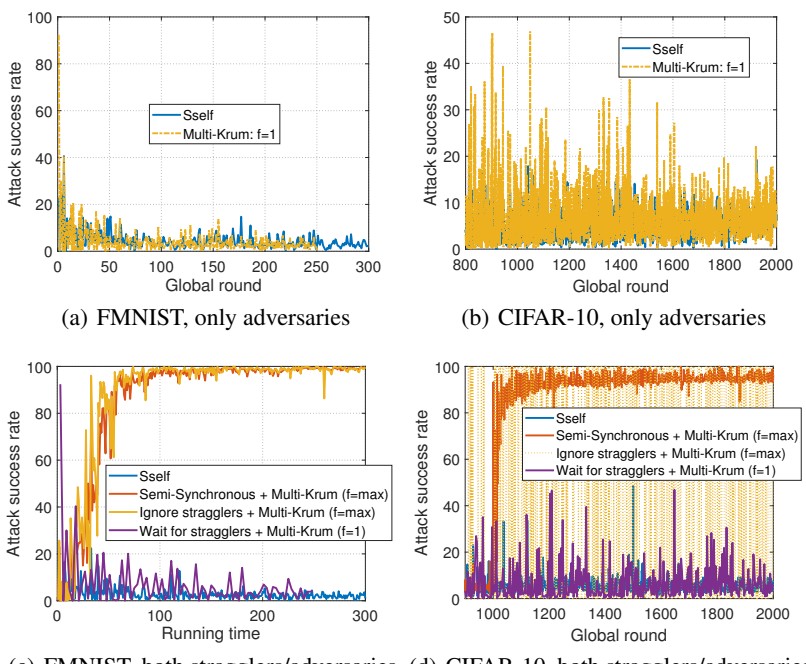

(a) FMNIST, only adversaries  (b) CIFAR-10, only adversaries

(c) FMNIST, both stragglers/adversaries  (d) CIFAR-10, both stragglers/adversaries

Figure C.3: Performance comparison with Multi-Krum under backdoor attack. We set $C = 0.1$ and $r = 0.1$.

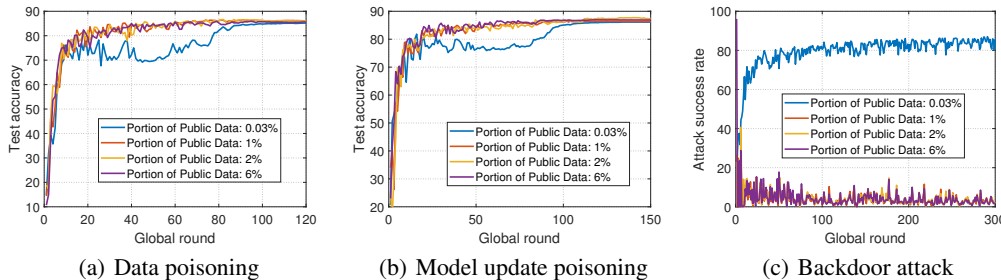

(a) Data poisoning      (b) Model update poisoning      (c) Backdoor attack

Figure D.1: Impact of portion of public data at the server using FMNIST. We set $C = 0.1$, $r = 0.1$ for the backdoor attack and $C = 0.2$, $r = 0.2$ for the others.

$f$ large enough. However, this scheme still suffers from the effect of stragglers, which significantly slows down the overall training process.

Fig. C.3 compares Sself with Multi-Krum under scaled backdoor attack. The results are consistent with the results in Fig. C.2, confirming the advantage of Sself over Multi-Krum combined with straggler-mitigating schemes.

## E    IMPACT OF PUBLIC DATA

In the experiments in our main manuscript, we utilized 2% of the training data samples as public data to defend against adversarial attacks. In this section, to observe the impact of the portion of public data, we performed additional experiments by changing the portion of public data under three attack scenarios in a synchronous setup. In the main manuscript, we let 2% of the entire training set to be the public data and the remaining data to be the training data at the devices for a fair comparison with other schemes. Here, the overall training set is utilized at the devices and among them, a certain portion of data are collected at the server. Fig. D.1 shows the results with various portions of public data on FMNIST. From the results, it can be seen that our Sself protects the system against adversarial attacks with only a small amount of public data. But as shown in the plot where the portion of the public data is 0.03%, if the amount of public data becomes smaller than a certain threshold, the robustness of Sself does suffer.

## F    EXPERIMENTS ON COVID-19 DATASET OPEN TO PUBLIC

In this section, we performed additional experiments on Kaggle's Covid-19 dataset[1] which is open to public. We consider both model update poisoning and data poisoning attacks in a synchronous setup. Image classification is performed to detect Covid-19 using Chest X-ray images. The dataset consists of 317 color images of $3480 \times 4248$ pixels in 3 classes (Normal, Covid and Viral-Pneumonia). There are 251 training images and 66 test images. We resized the images into $224 \times 224$ pixels and used convolutional neural network with 6 convolutional layers and 1 fully connected layer. We used 6% of the training data as the public data. We divided the remaining training samples into 10 devices and set $C = 1$ and $r = 0.1$.

Fig. F.1 shows the results of different schemes under data poisoning and model update poisoning attacks on Covid-19 dataset. As other baseline schemes, our Sself shows robustness against model update poisoning attack. In data poisoning attack, our Sself shows the best performance compared to other schemes. In conclusion, utilizing part of the open medical dataset as public data, we show that our Sself could effectively defend against model update and data poisoning attacks.

---

[1]https://www.kaggle.com/pranavraikokte/covid19-image-dataset

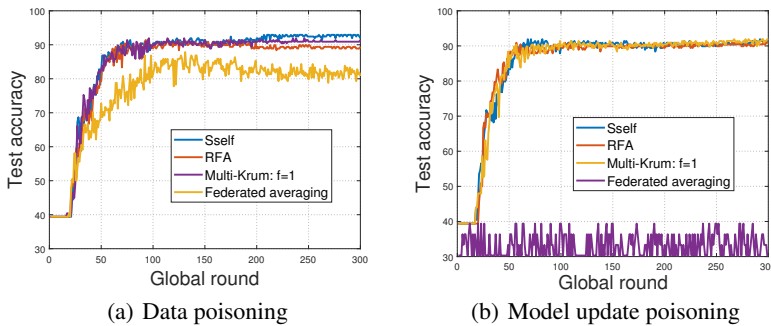

(a) Data poisoning  (b) Model update poisoning

Figure F.1: Performance of different schemes on medical dataset (Covid-19 image dataset) under data and model update poisoning attacks. We set $C = 1$, $r = 0.1$.

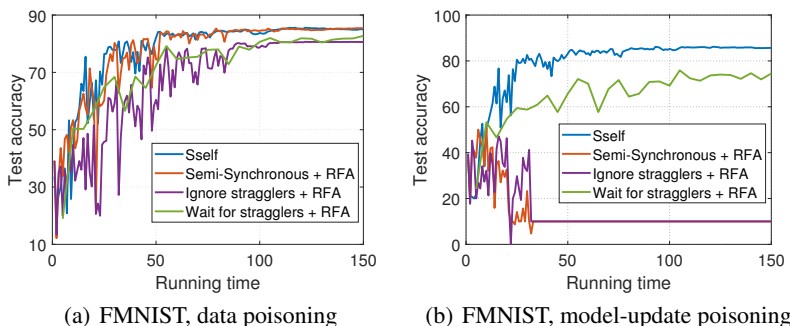

(a) FMNIST, data poisoning  (b) FMNIST, model-update poisoning

Figure G.1: Performance in a more severe straggler scenario where each device can have delay of $0$ to $4$. Data and model-update poisoning attacks are considered with FMNIST. We set $C = 0.4$, $r = 0.2$.

## G  EXPERIMENTS IN A MORE SEVERE STRAGGLER SCENARIO

When modeling stragglers, we gave a delay of $0$, $1$, $2$ to each device in the experiments of main manuscript. In this section, each device can have delay of $0$ to $4$ which is again determined independently and uniformly random. In Fig. G.1, we show the results with both stragglers and adversaries under data and model-update poisoning on FMNIST dataset. We set $C$ to $0.4$ and $r$ to $0.2$. It can be seen that our Sself still shows the best performance under both data poisoning and model-update poisoning compared to other baseline schemes.

## H  EXPERIMENTS WITH VARYING PORTION OF ADVERSARIES

In this section, we show the performance of Sself with varying portion of adversaries under data and model poisoning attacks. We do not consider stragglers here. We set $\delta$ to 1 and $E_{th}$ to 1 as in the experiments of the main manuscript. Fig. H.1 shows the results with different attack ratio on FMNIST dataset. For data poisoning, our Sself shows robustness against up to 0.4 of the attack ratio, but with 0.5 or higher, performance is degraded. For model update poisoning, it can be seen that our Sself performs well even with a higher attack ratio.

## I  PROOF OF THEOREM 1

### I.1  ADDITIONAL NOTATIONS FOR PROOF

After receiving the results at global round $t$, the server first performs entropy-based filtering and obtain $U_t^{(0)}$, $U_t^{(1)}$,..., $U_t^{(t)}$. Let $\mathbf{w}_t^j(k)$ be the model of the $k$-th benign device after $j$ local updates

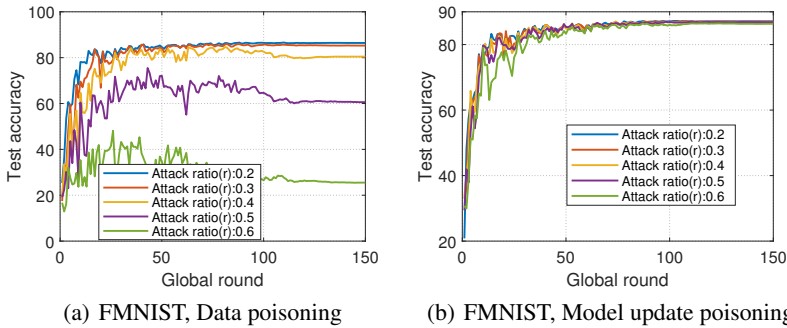

Figure H.1: Performance with varying portion of adversaries. Data and model-update poisoning attacks are considered with FMNIST. We set $C = 0.2$.

starting from global round $t$. At global round $t$, each device receives the current global model $\mathbf{w}_t$ and round index $t$ from the server, and sets its initial model to $\mathbf{w}_t$, i.e., $\mathbf{w}_t^0(k) \leftarrow \mathbf{w}_t$ for all $k = 1, \ldots, N$. Then each $k$-th benign device performs $E$ local updates of stochastic gradient descent (SGD) with learning rate $\eta$:

$$\mathbf{w}_t^j(k) \leftarrow \mathbf{w}_t^{j-1}(k) - \eta \nabla F_k(\mathbf{w}_t^{j-1}(k), \xi_t^{j-1}(k)) \; for \; j = 1, \ldots, E, \tag{8}$$

where $\xi_t^j(k)$ is a set of data samples that are randomly selected from the $k$-th device during the $j$-th local update at global round $t$. After $E$ local updates, the $k$-th benign device transmits $\mathbf{w}_t^E(k)$ to the server. However, in each round, the adversarial devices transmit poisoned model parameters.

Using these notations, the parameters defined in Section 2 can be rewritten as follows:

$$\mathbf{v}_{t+1}^{(i)} = \sum_{k \in U_t^{(i)}} \beta_i(k) \mathbf{w}_i^E(k) \; \text{where} \; \beta_i(k) \propto \frac{m_k}{\{F_{pub}(\mathbf{w}_i^E(k))\}^\delta} \; \text{and} \; \sum_{k \in U_t^{(i)}} \beta_i(k) = 1 \tag{9}$$

$$\mathbf{z}_{t+1} = \sum_{i=0}^{t} \alpha_t(i) \mathbf{v}_{t+1}^{(i)} \; \text{where} \; \alpha_t(i) \propto \frac{\sum_{k \in U_t^{(i)}} m_k}{(t-i+1)^c} \; \text{and} \; \sum_{i=0}^{t} \alpha_t(i) = 1 \tag{10}$$

$$\mathbf{w}_{t+1} = (1 - \gamma) \mathbf{w}_t + \gamma \mathbf{z}_{t+1} \tag{11}$$

## I.2 KEY LEMMA

We introduce the following key lemma for proving Theorem 1. Our proof is largely based on the convergence proof of FedAsync in (Xie et al., 2019a).

**Lemma 1** *Suppose Assumptions 1, 2 hold and the learning rate $\eta$ is set to be less than $\frac{1}{L}$. Consider the $k$-th benign device that received the current global model $\mathbf{w}_t$ from the server at global round $t$. After $E$ local updates, the following holds:*

$$\mathbb{E}[F(\mathbf{w}_t^E(k)) - F(\mathbf{w}^*)|\mathbf{w}_t^0(k)] \leq (1 - \eta\mu)^E [F(\mathbf{w}_t^0(k)) - F(\mathbf{w}^*)] + \frac{E\rho_1\eta}{2}. \tag{12}$$

*Proof of Lemma 1.* First, consider one step of SGD in the $k$-th local device. For a given $\mathbf{w}_t^{j-1}(k)$, for all global round $t$ and for all local updates $j \in \{1, \ldots, E\}$, we have

$$\mathbb{E}[F(\mathbf{w}_t^j(k)) - F(\mathbf{w}^*)|\mathbf{w}_t^{j-1}(k)]$$

$$\leq F(\mathbf{w}_t^{j-1}(k)) - F(\mathbf{w}^*) - \eta\mathbb{E}[\nabla F(\mathbf{w}_t^{j-1}(k))^T\nabla F_k(\mathbf{w}_t^{j-1}(k), \xi_t^{j-1}(k))|\mathbf{w}_t^{j-1}(k)]$$

$$+ \frac{L\eta^2}{2}\mathbb{E}[\|\nabla F_k(\mathbf{w}_t^{j-1}(k), \xi_t^{j-1})\|^2|\mathbf{w}_t^{j-1}(k)] \qquad \blacktriangleright \text{ SGD update and } L\text{-smoothness}$$

$$\leq F(\mathbf{w}_t^{j-1}(k)) - F(\mathbf{w}^*) + \frac{\eta}{2}\mathbb{E}[\|\nabla F(\mathbf{w}_t^{j-1}(k)) - \nabla F_k(\mathbf{w}_t^{j-1}(k), \xi_t^{j-1}(k))\|^2|\mathbf{w}_t^{j-1}(k)]$$

$$- \frac{\eta}{2}\|\nabla F(\mathbf{w}_t^{j-1}(k))\|^2 \qquad \blacktriangleright \eta < \frac{1}{L}$$

$$\leq F(\mathbf{w}_t^{j-1}(k)) - F(\mathbf{w}^*) - \frac{\eta}{2}\|\nabla F(\mathbf{w}_t^{j-1}(k))\|^2 + \frac{\eta\rho_1}{2} \qquad \blacktriangleright \text{ Assumption 2}$$

$$\leq (1 - \eta\mu)[F(\mathbf{w}_t^{j-1}(k)) - F(\mathbf{w}^*)] + \frac{\eta\rho_1}{2} \qquad \blacktriangleright \mu\text{-strongly convexity}$$

$$\tag{13}$$

Applying above result to $E$ local updates in $k$-th local device, we have

$$\mathbb{E}\left[F(\mathbf{w}_t^E(k)) - F(\mathbf{w}^*)|\mathbf{w}_t^0(k)\right]$$

$$= \mathbb{E}[\,\mathbb{E}[F(\mathbf{w}_t^E(k)) - F(\mathbf{w}^*)|\mathbf{w}_t^{E-1}(k)]|\mathbf{w}_t^0(k)\,] \qquad \blacktriangleright \text{ Law of total expectation}$$

$$\leq (1 - \eta\mu)\mathbb{E}[[F(\mathbf{w}_t^{E-1}(k)) - F(\mathbf{w}^*)]|\mathbf{w}_t^0(k)] + \frac{\eta\rho_1}{2} \qquad \blacktriangleright \text{ Inequality (13)}$$

$$\vdots$$

$$\leq (1 - \eta\mu)^E[F(\mathbf{w}_t^0(k)) - F(\mathbf{w}^*)] + \frac{\eta\rho_1}{2}\sum_{j=1}^{E}(1 - \eta\mu)^{j-1}$$

$$= (1 - \eta\mu)^E[F(\mathbf{w}_t^0(k)) - F(\mathbf{w}^*)] + \frac{\eta\rho_1}{2}\frac{1 - (1 - \eta\mu)^E}{\eta\mu} \qquad \blacktriangleright \text{ From } \eta < \frac{1}{L} \leq \frac{1}{\mu}, \ \eta\mu < 1$$

$$\leq (1 - \eta\mu)^E[F(\mathbf{w}_t^0(k)) - F(\mathbf{w}^*)] + \frac{E\eta\rho_1}{2} \qquad \blacktriangleright \text{ From } \eta\mu < 1, \ 1 - (1 - \eta\mu)^E \leq E\eta\mu$$

## I.3 PROOF OF THEOREM 1

Now utilizing Lemma 1, we provide the proof for Theorem 1. First, consider one round of global aggregation at the server. For a given $\mathbf{w}_{t-1}$, the server updates the global model according to equation

(11). Then for all $t \in 1, \ldots, T$, we have

$$
\mathbb{E}[F(\mathbf{w}_t) - F(\mathbf{w}^*)|\mathbf{w}_{t-1}]
$$

$$
\stackrel{(a)}{\leq} (1-\gamma)[F(\mathbf{w}_{t-1}) - F(\mathbf{w}^*)] + \gamma\mathbb{E}[F(\mathbf{z}_t) - F(\mathbf{w}^*)|\mathbf{w}_{t-1}]
$$

$$
\stackrel{(b)}{\leq} (1-\gamma)[F(\mathbf{w}_{t-1}) - F(\mathbf{w}^*)] + \gamma\sum_{i=0}^{t-1}\alpha_{t-1}(i)\mathbb{E}[F(\mathbf{v}_t^i) - F(\mathbf{w}^*)|\mathbf{w}_{t-1}]
$$

$$
\stackrel{(c)}{\leq} (1-\gamma)[F(\mathbf{w}_{t-1}) - F(\mathbf{w}^*)] + \gamma\sum_{i=0}^{t-1}\alpha_{t-1}(i)\sum_{k\in U_{t-1}^{(i)}}\beta_i(k)\mathbb{E}[F(\mathbf{w}_i^E(k)) - F(\mathbf{w}^*)|\mathbf{w}_{t-1}]
$$

$$
= (1-\gamma)[F(\mathbf{w}_{t-1}) - F(\mathbf{w}^*)] + \gamma\sum_{i=0}^{t-1}\alpha_{t-1}(i)\Big\{\sum_{k\in B_{t-1}^{(i)}}\beta_i(k)\mathbb{E}[F(\mathbf{w}_i^E(k)) - F(\mathbf{w}^*)|\mathbf{w}_{t-1}]
$$

$$
+ \sum_{k\in M_{t-1}^{(i)}}\beta_i(k)\mathbb{E}[F(\mathbf{w}_i^E(k)) - F(\mathbf{w}^*)|\mathbf{w}_{t-1}]\Big\}
$$

$$
\stackrel{(d)}{\leq} (1-\gamma + \gamma\alpha_{t-1}(t-1)(1-\Omega_{t-1}^{t-1})(1-\eta\mu)^E)[F(\mathbf{w}_{t-1}) - F(\mathbf{w}^*)] + \frac{E\eta\rho_1\gamma}{2}
$$

$$
+ \gamma(1-\eta\mu)^E\sum_{i=0}^{t-2}\alpha_{t-1}(i)\sum_{k\in B_{t-1}^{(i)}}\beta_i(k)\left[F(\mathbf{w}_i^0(k)) - F(\mathbf{w}^*)\right]
$$

$$
+ \gamma\sum_{i=0}^{t-1}\alpha_{t-1}(i)\sum_{k\in M_{t-1}^{(i)}}\beta_i(k)\mathbb{E}[F(\mathbf{w}_i^E(k)) - F(\mathbf{w}^*)|\mathbf{w}_{t-1}]
$$

$$
\stackrel{(e)}{\leq} (1-\gamma + \gamma\alpha_{t-1}(t-1)(1-\Omega_{t-1}^{t-1})(1-\eta\mu)^E)[F(\mathbf{w}_{t-1}) - F(\mathbf{w}^*)] + \gamma\Omega_{max}\Gamma
$$

$$
+ \gamma(1-\eta\mu)^E\sum_{i=0}^{t-2}\alpha_{t-1}(i)\sum_{k\in B_{t-1}^{(i)}}\beta_i(k)\left[F(\mathbf{w}_i^0(k)) - F(\mathbf{w}^*)\right] + \frac{E\eta\rho_1\gamma}{2}
$$

$$
\stackrel{(f)}{\leq} (1-\gamma + \gamma\alpha_{t-1}(t-1)(1-\Omega_{t-1}^{t-1})(1-\eta\mu)^E)[F(\mathbf{w}_{t-1}) - F(\mathbf{w}^*)] + \gamma\Omega_{max}\Gamma
$$

$$
+ \gamma(1-\eta\mu)^E\sum_{i=0}^{t-2}\alpha_{t-1}(i)\sum_{k\in B_{t-1}^{(i)}}\beta_i(k)\frac{1}{2\mu}\|\nabla F(\mathbf{w}_i^0(k))\|^2 + \frac{E\eta\rho_1\gamma}{2}
$$

$$
\stackrel{(g)}{\leq} (1-\gamma + \gamma\alpha_{t-1}(t-1)(1-\Omega_{t-1}^{t-1})(1-\eta\mu)^E)[F(\mathbf{w}_{t-1}) - F(\mathbf{w}^*)] + \gamma\Omega_{max}\Gamma
$$

$$
+ \frac{E\eta\rho_1\gamma}{2} + \frac{\gamma(1-\alpha_{t-1}(t-1))(1-\eta\mu)^E\rho_2}{2\mu}
$$

$$
\stackrel{(h)}{\leq} (1-\gamma + \gamma\alpha_{t-1}(t-1)(1-\Omega_{t-1}^{t-1})(1-\eta\mu)^E)[F(\mathbf{w}_{t-1}) - F(\mathbf{w}^*)]
$$

$$
+ \frac{\gamma(E\rho_1 + (1-\alpha_{t-1}(t-1))\rho_2 + 2\mu\Omega_{max}\Gamma)}{2\mu} \tag{14}
$$

where $(a)$, $(b)$, $(c)$ come from convexity, $(d)$ follows Lemma 1, $(e)$ comes from $\Omega_{max} = \max\limits_{0\leq i\leq t, 0\leq t\leq T}\Omega_t^{(i)}$ and the Assumption 4. $(f)$ is due to $\mu$-strongly convexity, $(g)$ is from Assumption 3 and $(h)$ comes from $\eta\mu < 1$. Note that $\sum_{i=0}^{t-1}\alpha_{t-1}(i) = 1$ for all $t$.

Applying the above result to $T$ global aggregations in the server, we have

$$\mathbb{E}[F(\mathbf{w}_T) - F(\mathbf{w}^*)|\mathbf{w}_0]$$

$$\overset{(a)}{=} \mathbb{E}\left[\mathbb{E}[F(\mathbf{w}_T) - F(\mathbf{w}^*)|\mathbf{w}_{T-1}]|\mathbf{w}_0\right]$$

$$\overset{(b)}{\leq} \mathbb{E}\left[(1 - \gamma + \gamma\alpha_{T-1}(T-1)(1 - \Omega_{T-1}^{T-1})(1 - \eta\mu)^E)[F(\mathbf{w}_{T-1}) - F(\mathbf{w}^*)]|\mathbf{w}_0\right.$$
$$\left. + \frac{\gamma(E\rho_1 + (1 - \alpha_{t-1}(t-1))\rho_2 + 2\mu\Omega_{max}\Gamma)}{2\mu}\right]$$

$$\overset{(c)}{\leq} \prod_{\tau=0}^{T-1}(1 - \gamma + \gamma\alpha_\tau(\tau)(1 - \Omega_\tau^\tau)(1 - \eta\mu)^E)[F(\mathbf{w}_0) - F(\mathbf{w}^*)] + \frac{\gamma(E\rho_1 + (1 - \alpha_{T-1}(T-1))\rho_2 + 2\mu\Omega_{max}\Gamma)}{2\mu}$$

$$+ \sum_{\tau=1}^{T-1}\frac{\gamma(E\rho_1 + (1 - \alpha_{T-1-\tau}(T-1-\tau))\rho_2 + 2\Omega_{max}\Gamma)}{2\mu}\prod_{k=1}^{\tau}(1 - \gamma + \gamma\alpha_{T-k}(T-k)(1 - \Omega_{T-k}^{T-k})(1 - \eta\mu)^E)$$

$$\overset{(d)}{\leq} (1 - \gamma + \gamma(1 - \eta\mu)^E)^T[F(\mathbf{w}_0) - F(\mathbf{w}^*)] + \left[1 - \{1 - \gamma + \gamma(1 - \eta\mu)^E\}^T\right]\frac{E\rho_1 + \rho_2 + 2\mu\Omega_{max}\Gamma}{2\mu(1 - (1 - \eta\mu)^E)}$$

$$\overset{(e)}{\leq} (1 - \gamma + \gamma(1 - \eta\mu)^E)^T[F(\mathbf{w}_0) - F(\mathbf{w}^*)] + \left[1 - \{1 - \gamma + \gamma(1 - \eta\mu)^E\}^T\right]\frac{\rho_1 + \rho_2 + 2\mu\Omega_{max}\Gamma}{2\eta\mu^2}$$

$$= \nu^T[F(\mathbf{w}_0) - F(\mathbf{w}^*)] + (1 - \nu^T)C'$$

which completes the proof. Here, $(a)$ comes from the *Law of total expectation*, $(b)$, $(c)$ are due to inequality (14). And $(d)$ comes from $0 \leq \alpha_t(i) \leq 1$ and $0 \leq \Omega_t^{(i)} < 1$ for all $i$, $t$. In addition, $(e)$ is from $\eta\mu \leq 1$ and $E$ is a positive integer.

