# OpenReview forum: "Sself: Robust Federated Learning against Stragglers and Adversaries"
_ICLR.cc/2021/Conference — Reject_

### Official Review · AnonReviewer3 · 2020-10-24
**Interesting idea with unfair baselines and unclear details.**

**Rating:** 4
**Confidence:** 3

**Review:**

Review: This paper proposes Sself to achieve robustness against adversary when there are straggler.
They uses semi-synchronous scheme to handle the straggler. In each round, the server use entropy based filtering to filter models with large entropy. Then the models are aggregated based on their weights and staleness. The main contribution of this paper is to propose an asynchronous training scheme in the presence of Byzantine workers.

=======================================================

Pros:

- Using semi-synchronous scheme to handle straggler is interesting.

- The entropy based filtering and loss based averaging looks interesting.

- The`experiments presented in the paper look good.

=======================================================

Cons:

- The computation cost in each round grows linearly with time.
- Unfair baselines:
  - The comparison with [1] is unfair because they don't assume the server to have training data. Instead, Sself should be compared with Zeno++[2] which has similar setting;
  - `Waiting for straggler` is not a good baseline. To handle the straggler, [3] suggest the server can simply send requests to 30 percent more workers for updates and wait for the fastest replies.
- Assuming public and IID data on the server is a very strong assumption. The data collection schemes do not consider the existence of Byzantine worker.
- It is not clear to me how this method can defend backdoor attack. In [4], they mentioned that the goal is to achieve high accuracy on main task and an attacker-chosen subtask. So it should behave good on the main task, especially comparing to the models trained on non-i.i.d. good workers. Is it possible to improve the backdoor attack such that it can deceive Sself?

- It is not very convincing to me that the filtering threshold is easy to select. The entropy of the models decrease over time, does the threshold change as well? Is the threshold a prior knowledge or learned from data?

- What are the numbers of adversaries can be tolerated by this method?

=======================================================

Minor comments:

- The curves are a bit bumpy.


[1] Pillutla, Krishna, Sham M. Kakade, and Zaid Harchaoui. "Robust aggregation for federated learning." arXiv preprint arXiv:1912.13445 (2019).

[2] Xie, Cong, Sanmi Koyejo, and Indranil Gupta. "Zeno++: Robust Fully Asynchronous SGD." arXiv preprint arXiv:1903.07020 (2019).

[3] Bonawitz, Keith, et al. "Towards federated learning at scale: System design." arXiv preprint arXiv:1902.01046 (2019).

[4] Bagdasaryan, Eugene, et al. "How to backdoor federated learning." International Conference on Artificial Intelligence and Statistics. PMLR, 2020.

---

> ### Author Response · Authors · 2020-11-17
> **Response to Reviewer 3**
>
> $\textbf{Comment on the computational cost:}$ We first note that the computational cost depends on the number of received results at the server in each global round, and does not increase linearly in time. Suppose $K$ devices are selected to participate at each global round $t$. When there are no stragglers, the server processes all the results of $K$ devices at each round. If stragglers exist, the selected $K$ devices in round $t$ successfully send their results to the server at the following global rounds: ${t, t+1, t+2, t+3…}$. Assuming a specific probability distribution for the arrival of results, the expected number of results at each global round does not exceed $K$. Hence, the expected computational cost is the same as the one with no stragglers and obviously does not increase linearly in time.
>
> $\textbf{Fair comparison with RFA:}$ To make a fair comparison, in Sself, we let 2% of the training set of each dataset (MNIST, FMNIST, CIFAR-10) to be the public data, and the remaining 98% be the data of each device. For other schemes we let 100% of training data of each dataset be at the devices. Hence, the total amount of data utilized in the system is set to be identical compared to other schemes including RFA.
>
> $\textbf{Comparison with Zeno:}$ We note that although the authors of (Xie et al.(2019)) considered both stragglers and adversaries, they do not target non-IID federated learning but targets IID distributed learning setup. Thanks to the reviewer’s comment, we clearly mentioned this difference in our revised manuscript. We also confirmed by experiments that the scheme of (Xie et al.(2019)) suffers from both stragglers/adversaries, because of the non-IID data distribution in a federated learning setup that we are targeting. As an example, in model-update poisoning attack with stragglers, Zeno++ achieves accuracy of 27.2% while Sself achieves accuracy of 85.7%.
>
> $\textbf{Straggler mitigating scheme:}$ The scheme that reviewer suggested can be viewed as a kind of “ignore stragglers” scheme, but requires more devices (along with more communication and computation resources) at each global round than our scheme and other baselines. A fairer version of “ignore stragglers” scheme is already in our original manuscript: the server sends requests to the same number of devices as other baselines and ignore the results of the slow devices. Regarding the “wait for stragglers” scheme, we will keep the results of this method in our manuscript to provide insights especially when combined with the robust schemes against adversaries (see Fig. 4 for the details).
>
> $\textbf{Public data:}$ Actually, there are various types of public data widely open to public (e.g., medical data). Moreover, data centers generally have public data collected from a group of trusted third party or a group of trusted workers, as also described in Zeno++ paper that the reviewer suggested and other federated learning papers such as [Li 2020].
>
> [Li 2020] Qinbin Li, Bingsheng He, Dawn Song, "Model-Agnostic Round-Optimal Federated Learning via Knowledge Transfer." arXiv preprint arXiv:2010.01017 (2020).
>
> $\textbf{Backdoor attack:}$ Note that we considered two kinds of backdoor attacks: scaled version and no-scaled version. In scaled backdoor attack, the attack has similar effect as model update poisoning so the models from the adversaries can be easily filtered out by entropy-filtering. Regarding the no-scaled backdoor attack, although the attack success rate increases as global round grows, the effect of adversaries can be suppressed for a while by loss-weighted averaging as can be seen from the results in the supplementary material.
>
> $\textbf{Entropy-based filtering:}$ The entropy of models from adversaries do not decrease over time, and has significantly larger entropies compared to the benign devices. Hence, Sself works well even with a naively chosen filtering threshold, as already shown in the experiments in supplementary material Section C with various Eth values. Moreover, even when this method does not filter out the results of adversaries successfully, the loss-weighted averaging method can reduce the impact of remaining adversaries. The threshold value can be learned from data, but the same set of Eth values works well for both FMNIST and CIFAR-10 as can be seen in the plots in the supplementary material.
>
> $\textbf{Number of adversaries that can be tolerated:}$ As in the RFA or other median-based methods, there is no specific theoretical bound on the number of adversaries that our method can tolerate. It depends on the size of the system and the portion of stragglers. In supplementary material, we provided additional experimental results with varying portions of adversaries. As for other median based schemes, the performance gradually degrades as the portion of adversaries increases.

---

### Official Review · AnonReviewer1 · 2020-10-28
**Good algorithm, but I have some concerns regarding the theory**

**Rating:** 5
**Confidence:** 3

**Review:**

**Paper summary**

The paper claims to propose the first algorithm that can handle adversarial machines and stragglers simultaneously in the federated learning setting. To handle stragglers, the paper takes a semi-synchronous approach by taking a weighted sum of gradients depending on staleness. To handle adversarial machines, the algorithm uses an entropy based filtering and a loss based averaging strategy. Note that to handle the adversaries, the algorithm needs a public dataset at the server, using which it can evaluate the entropy and loss scores of each gradient.

**Strengths**
1. The problem of handling stragglers and adversarial machines (including data poisoning adversaries) seems a very relevant problem. The paper claims to be the first to solve this (however I think Xie et al.(2019) also solve a similar problem).
2. The handling of stragglers seems to be theoretically backed (although I have some concerns about the handling of adversarial machines) and Theorem 1 shows that the convergence up to some error is fast.
3. The experiments show that the algorithm beats or matches existing algorithms on MNIST, CIFAR10 etc.


**Concerns**
1. In the proof of Theorem 1, I could not find the analysis of the Entropy filtering step of the algorithm.
2. I think Lemma 1 is only applicable on the non-adversarial machines. In the proof of Theorem 1, inequality (d) (on page 18) applies Lemma 1 on all the machines including the adversarial ones.

For these reasons, I think Theorem 1 does not give correct guarantees for the algorithm.

**Score justification**

As mentioned in the Concerns section, I am not sure if the theoretical guarantees for the algorithm are correct.

**References**

Xie, C., Koyejo, O. and Gupta, I., 2019. Zeno++: Robust Fully Asynchronous SGD.

---

> ### Author Response · Authors · 2020-11-17
> **Response to Reviewer 1**
>
> We would like to note that this is the first work to handle stragglers/adversaries at the same time, in a federated learning setup with non-IID data across the devices. Although the authors of (Xie et al.(2019)) consider both stragglers and adversaries, they do not target non-IID federated learning but targets IID distributed learning setup. Thanks to the reviewer’s comment, we clarified this difference in our revised manuscript. We also confirmed by experiments that the scheme of (Xie et al.(2019)) suffers from both stragglers/adversaries, because of the non-IID data distribution in a federated learning setup that we are targeting. As an example, in model-update poisoning attack with stragglers, Zeno++ achieves accuracy of 27.2% while Sself achieves accuracy of 85.7%.
>
> $\textbf{Entropy-based filtering in Theorem 1:}$ Actually, the entropy-based filtering step is already reflected in the proof of Theorem 1. The set U_t^((i)) in the proof of Theorem 1 can be viewed as a set after the entropy filtering process, as in Line 6 of Algorithm 1. We added this description to make the point clearer in the proof of Theorem 1.
>
> $\textbf{Comment on Theorem 1:}$ We thank the reviewer for the comment. We modified the theoretical bound by considering the effect of adversaries. Only the models from the benign devices satisfy Lemma 1. For adversarial devices, we have an additional error term. Please refer to our revised manuscript for the details.

---

### Official Review · AnonReviewer4 · 2020-10-29
**Official Blind Review #4**

**Rating:** 4
**Confidence:** 4

**Review:**

This paper considers federated learning with straggling and adversarial devices. To tackle stragglers, the paper proposes semi-synchronous averaging wherein models with the same staleness are first averaged together, and then a weighted average of the results with different stateless is computed. To mitigate adversaries, the paper proposes to first perform entropy-based filtering to remove suspected outliers, and then compute loss-weighted average. The server is assumed to have some public data, which is used for entropy-based filtering. Together, the proposed algorithm is called semi-synchronous entropy and loss based filtering (Sself).

Strong points:

1. Mitigating adversarial attacks, especially model poisoning and backdoor attacks, is an important challenge in federated learning.

2. The proposed algorithm is simple. The paper is well-written, and easy to follow.

Weak points:

1. Theorem 1 does not seem to consider adversarial devices, and the proof seems to only the semi-synchronous part of Sself without entropy-based filtering and loss-weighted average. Intuitively, the convergence performance should degrade with the number of adversarial devices. However, the theorem statement does not seem to indicate so. If the theorem only analyzes the semi-synchronous part of Sself, then this should be explicitly mentioned. The theorem in its current form is a bit misleading.

Further, even when there are no adversaries, what is the theoretical improvement over FedAsync? Specifically, what is the impact of staleness on convergence. It is important to add remarks to elaborate the gains qualitatively.

2. Experiments are performed for the simplistic case when each device has a delay of 0, 1, or 2 rounds (chosen uniformly at random and independently for each device). Further, experiments (in the main body of the paper and supplementary material) are for a limited number of adversarial attacks. The performance can be evaluated for practically motivated straggler models and more powerful known attacks. For instance, the attack from the following paper:

G. Baruch, M. Baruch, Y. Goldberg, “A Little Is Enough: Circumventing Defenses For Distributed Learning”, NeurIPS 2019.

While proposing entropy-based filtering, the authors hypothesize that “if the local model is poisoned, e.g., by reverse sign attack, the model is more likely to predict randomly for all classes and thus has a high entropy”. However, some attacks such as targeted backdoor attacks typically do not hurt overall accuracy. So, it is not clear why the poisoned model is likely to predict randomly for all classes. It will be helpful to add more evidence.

3. Evaluating the loss on the public data for each device may incur significant computational complexity. It is important to elaborate on how much the complexity at the server will increase by using entropy-based filtering on public data.

Other suggestions:

1. Fig. 2, should the x-axis label be round number than running time?

2. In experiments, can you quantify the performance gain of Sself over FedAsync? For instance, in Fig. 2, FedAsnc seems to be almost on par with Sself. It will be helpful to quantify the performance improvement.

3. Zeno proposed in the following paper also uses public data at the server. If possible, it will be good to compare against Zeno for fairness.

Cong Xie, Sanmi Koyejo, Indranil Gupta, “Zeno: Distributed Stochastic Gradient Descent with Suspicion-based Fault-tolerance”, ICML 2019.

In summary, the theoretical result (Theorem 1) is weak as it does not seem to consider adversarial devices. In experiments, the stragglers are simulated in a simplistic manner, and the gains over FedAsync are not quantified. It will be good to consider the weak points and suggestions to improve the paper. Currently, the novelty seems to be fairly limited.

-------------- Post-Rebuttal Comments -----------------
Thanks to the authors for their response, and for updating the manuscript. Some of my queries were clarified. However, updated Theorem 1 seems to raise more questions. In particular, Assumption 4 looks very restrictive to me. If adversaries manage to produce large values of \Gamma, they can inflict a large error as per (7). The paragraph after Theorem 1 does not mention how entropy and loss based filtering methods can achieve small \Omega_{max}, but only says that "if the entropy-based filtering method successfully filters out the model poisoned devices, and the loss-weights \beta_i(k) of the adversaries are significantly small for data poisoning and backdoor attacks... ... then we have a small error term". It is not clear what guarantees the filtering schemes yield. Due to these reasons, I still think the paper is not yet ready for publication.

---

> ### Author Response · Authors · 2020-11-17
> **Response to Reviewer 4**
>
> $\textbf{Comment on Theorem 1:}$ We thank the reviewer for this comment. We modified the theoretical bound to reflect the effect of adversaries, as we remarked in our response to Reviewer 2 above.
>
> $\textbf{Comparison with FedAsync:}$ When only considering the stragglers, the theoretical convergence speed is identical to the one with FedAsync. In practical scenarios with both stragglers and adversaries, although there are no theoretical results on FedAsync, experimental results in Fig. 4 show that it is challenging for FedAsync to handle both stragglers and adversaries simultaneously when combined with our entropy-based filtering and loss-weighted averaging methods. It is also impossible to combine FedAsync with Krum or median-based aggregation rules such as RFA.
>
> $\textbf{Experiment:}$ In supplementary material, we provided additional experimental results on delay scenario with more severe stragglers. The delay can be now 0,1,2,3,4 global rounds. The overall results still confirm the advantage of Sself in this scenario. Regarding the attack, the paper the reviewer suggested proposes a type of backdoor attack but is applicable in IID settings for distributed learning. Since we consider a non-IID federated learning scenario, we did not consider the attack of this paper. We think that we handled various attack scenarios including model-update poisoning, data poisoning and backdoor attacks in our original manuscript.
>
> $\textbf{Entropy-based filtering and backdoor attack:}$ As we stated in our manuscript, entropy-based filtering is utilized to combat model update poisoning, and loss-weighted averaging is used to combat data poisoning and backdoor attacks. In scaled backdoor attack, the attack has similar effect as model update poisoning so the models from the adversaries can be easily filtered out by entropy-filtering. Regarding the no-scaled backdoor attack, although the attack success rate increases as global round grows, the effect of adversaries can be suppressed for a while by loss-weighted averaging (not entropy-based filtering) as can be seen from the results in the supplementary material.
>
> $\textbf{Computational complexity:}$ We thank the reviewer for this comment. The computational complexity of Sself depends on the number of received models at each global round, and the running time for computing the entropy/loss with each model. Although direct comparison with other baselines is tricky, if we assume that the complexity of computing entropy or loss is linear in the number of model parameters as in the work of Zeno [1], Sself has larger complexity than that of RFA by a factor of $n_{pub}$ but still has smaller complexity compared to Krum. The additional computational complexity of Sself compared to RFA is the cost for better robustness against adversaries. We added the overall discussion in the revised manuscript.
>
> $\textbf{Regarding the x-axis:}$ Due to the stragglers, note that the running time of each global round is different across the schemes. The x-axis label should be the running time, not the global round.
>
> $\textbf{Performance comparison with FedAsync:}$ If we only have stragglers, we agree that both Sself and FedAsync perform well as in Fig. 2. The main results of the experiment section is Fig. 4, a scenario with both stragglers and adversaries. As we stated in the original manuscript, FedAsync is potentially hard to be implemented in conjunction with the robust methods against adversaries. For example, since the model is aggregated one-by-one, it is impossible to combine FedAsync with Krum or median-based aggregation rules such as RFA. It can be seen from Fig. 4 that it is challenging for FedAsync to handle both stragglers and adversaries simultaneously even when combined with our entropy-based filtering and loss-weighted averaging methods.
>
> $\textbf{Comparison with Zeno:}$ We first note that Zeno [1] is a scheme that considers only adversaries (not stragglers). Zeno++ [2] is an advanced version of Zeno, which tackles both stragglers and adversaries at the same time. The difference between our scheme and Zeno++ can be clearly described as follows. Although the authors of Zeno++ considered both stragglers and adversaries with the public data, they do not target non-IID federated learning but targets IID distributed learning setup. We confirmed by experiments that the scheme of Zeno++ suffers from both stragglers/adversaries, because of the non-IID data distribution in the federated learning setup that we are targeting. As an example, in model-update poisoning attack with stragglers, Zeno++[2] achieves accuracy of 27.2% while Sself achieves accuracy of 85.7%. Thanks to the reviewer’s comment, we clarified this difference in our revised manuscript.
>
> [1] Xie et al., "Zeno: Distributed stochastic gradient descent with suspicion-based fault-tolerance." ICML 2019.
>
> [2] Xie et al., "Zeno++: Robust Fully Asynchronous SGD." arXiv preprint arXiv:1903.07020 (2019).

---

### Official Review · AnonReviewer2 · 2020-11-02
**Recommendation to Reject**

**Rating:** 4
**Confidence:** 3

**Review:**

Summary:
The paper studies the problem of asynchronous training  with robustness to adversaries in federated learning. The authors propose an idea based on entropy based filtering to simultaneously filter out adversaries, and a weighted averaging technique to handle staleness in the gradients.

Pros:
-->I like the idea of using stale gradients that arrive past their due date in a delayed manner with appropriate weight.
--> The idea of using a public data set, and metrics from that data set to aide filtering out adversaries is interesting, and possibly worthy of pursuit.

Cons/Concerns:
--> The paper's analysis assumes that the entropy based filtering technique automatically filters out gradients from adversaries. In fact, this seems to play no role in the the analysis in Theorem 1 (Appendix G.1). In fact, this theorem proof seems to be identical to a proof that would exist if there were no adversaries (unless I missed something).

--> I find Theorem 1's result a bit confusing. As per the paper, they seem to be allowing an arbitrary number of stragglers (and arbitrary degree of straggling). For instance, for some value of t, it is possible that $S_t = U_\infty^{t},$ which means that none of the nodes selected in time $t$ ever send their updates. The theorem's assumptions do not seem to preclude this, but somehow, the result is blind to such straggling. Put differently, we expect that the performance must closely depend on the degree of asynchrony. However, this is not sufficiently captured in the statement of Theorem 2.

--> Overall, I find that the novelty is relatively limited, and the ideas seem to be a simple combination of those that have appeared in Zhao et. al 2018, and Xie et. al 2019

---

> ### Author Response · Authors · 2020-11-17
> **Response to Reviewer 2**
>
> $\textbf{Comment on Theorem 1:}$  We thank the reviewer for pointing this out. We modified the theoretical bound to reflect the effect of adversaries. For adversarial devices, we need an additional error term, as clarified in the revised manuscript.
>
> $\textbf{Straggler scenario:}$ If $U_t^{(t)} \neq \emptyset$ holds, then our result on Theorem 1 captures all straggling scenarios. This is a very weak assumption because we only need at least one device to successfully send its result to the server in the current episode t. Since the straggler contacts hundreds or more devices in each global round, we always can meet this condition in practice. We clarified this condition in the revised manuscript.
>
> $\textbf{Novelty compared with  (Zhao et. al 2018) and (Xie et. al 2019):}$  Regarding the novelty, we emphasize that the ideas of (Zhao et. al 2018) and (Xie et. al 2019) have totally different concepts compared to ours. First of all, in (Zhao et. al 2018), the server sends the public data to the devices to relieve the non-IID issue. Stragglers and adversaries are not considered at all in this previous work. In our work, we utilize the public data to perform entropy-based filtering and loss-weighted averaging to combat adversaries, a completely different concept and an orthogonal goal relative to the paper of (Zhao et. al 2018). Secondly, as we already stated in the Introduction, the authors of (Xie et. al 2019) propose an asynchronous scheme that is potentially hard to be implemented in conjunction with the robust methods against adversaries (i.e., median-based schemes). Instead of updating the model one-by-one as in (Xie et. al 2019), we propose a semi-synchronous scheme by taking a judicious mix of both synchronous and asynchronous approaches to handle stragglers more efficiently, and to be effectively combined with the robust schemes against adversaries.

---

### Author Response · Authors · 2020-11-17
**General comments to reviewers**

We would like to thank the reviewers for their efforts and constructive suggestions, which have greatly helped us to improve the paper. We have worked them into the revised version of the paper. The main changes are as follows:

1. We modified the theoretical bound in Theorem 1 to reflect the effect of adversaries.
2. We clarified the difference between the proposed Sself and Zeno++ in the revised manuscript.
3. We provided additional experimental results 1) in a more severe straggler scenario and 2) with varying portion of adversaries, in the supplementary material.

All the changes in the revised paper are marked in blue color. For more details, please refer to our official comments corresponding to each review.

---

### Decision · Program_Chairs · 2021-01-07
**Final Decision**

**Decision:**

Reject

**Comment:**

This paper proposes a combined method to address stragglers and adversaries in federated learning. Stragglers are overcome by allowing staleness in model aggregation. Adversaries are handled by using a public dataset to identify poisoned devices and adjusting their weights when doing model aggregation. However, the reviewers raised concerns about:
* The correctness of Theorem 1
* The novelty of the paper given that there has been significant previous work on straggler mitigation and robust aggregation in distributed learning.

As a result, I am unable to recommend the acceptance of the paper. However, the idea is certainly promising, and if built upon more rigorously can result in a nice and impactful paper. I hope that the authors can take the reviewers' feedback into account when revising the paper for a future submission!